# Lercanidipine Synergistically Enhances Bortezomib Cytotoxicity in Cancer Cells via Enhanced Endoplasmic Reticulum Stress and Mitochondrial Ca^2+^ Overload

**DOI:** 10.3390/ijms20246112

**Published:** 2019-12-04

**Authors:** A Reum Lee, Min Ji Seo, Jin Kim, Dong Min Lee, In Young Kim, Mi Jin Yoon, Hur Hoon, Kyeong Sook Choi

**Affiliations:** 1Department of Biochemistry and Molecular Biology, Ajou University, Suwon 16499, Korea; aremshy@ajou.ac.kr (A.R.L.); hisseomin22@ajou.ac.kr (M.J.S.); rlawls0297@ajou.ac.kr (J.K.); ldonglminl@ajou.ac.kr (D.M.L.); adela@ajou.ac.kr (I.Y.K.); seanara@ajou.ac.kr (M.J.Y.); 2Department of Biomedical Science, Ajou University Graduate School of Medicine, Suwon 16499, Korea; hhcmc75@ajou.ac.kr; 3Department of Surgery, Ajou University School of Medicine, Suwon 16499, Korea

**Keywords:** bortezomib, lercanidipine, ER stress, mitochondrial Ca^2+^ overload, paraptosis

## Abstract

The proteasome inhibitor (PI), bortezomib (Btz), is effective in treating multiple myeloma and mantle cell lymphoma, but not solid tumors. In this study, we show for the first time that lercanidipine (Ler), an antihypertensive drug, enhances the cytotoxicity of various PIs, including Btz, carfilzomib, and ixazomib, in many solid tumor cell lines by inducing paraptosis, which is accompanied by severe vacuolation derived from the endoplasmic reticulum (ER) and mitochondria. We found that Ler potentiates Btz-mediated ER stress and ER dilation, possibly due to misfolded protein accumulation, in MDA-MB 435S cells. In addition, the combination of Btz and Ler triggers mitochondrial Ca^2+^ overload, critically contributing to mitochondrial dilation and subsequent paraptotic events, including mitochondrial membrane potential loss and ER dilation. Taken together, our results suggest that a combined regimen of PI and Ler may effectively kill cancer cells via structural and functional perturbations of the ER and mitochondria.

## 1. Introduction 

Proteasome inhibition is an established treatment strategy for patients with multiple myeloma (MM) [1]. Since the introduction of the first-in-class proteasome inhibitor (PI), bortezomib (Btz), the therapeutic landscape for MM has seen the development of next-generation PIs (e.g., carfilzomib (Cfz) and ixazomib (Ixz)) and new classes of agents [1,2]. However, despite the success of PIs in treating hematological malignancies, including MM, the PIs have shown limited clinical efficacy as mono-treatments for solid tumors [3,4]. In addition, both innate and acquired resistance mechanisms can increasingly compromise the effectiveness of PI therapy [5]. Various mechanisms have been proposed as underlying PI resistance in solid tumors, including upregulated activity and increased subunit expression of proteasomes, proteasome β5-subunit mutations, protective autophagy, apoptosis-mediated resistance due to alteration of the Mcl-1/Noxa balance, elevation of P-glycoprotein resulting in Cfz efflux, and KRAS mutation associated with reprogramming of metabolic pathways [6]. As simply elevating the PI concentration may cause side effects, we must identify agents capable of sensitizing resistant cancer cells to the anti-cancer effects of PIs, if we hope to improve the effectiveness of PI-based cancer therapy. Recently, the repurposing of existing drugs for oncological indications has attracted attention in the development of cancer therapeutics, because it provides opportunities to expedite the availability of drugs to treat cancers in a cost-effective manner [7]. In this study, we show for the first time that Lercanidipine (Ler), a third-generation 1,4-dihydropyridine (DHP) known as a calcium channel blocker [8], can synergistically enhance the cytotoxicity of PIs in various types of cancer cells. Ler contributes to peripheral vasodilation by preventing Ca^2+^ ions from passing through L-type calcium channels in the cell membrane [9] and has shown impressive efficacy in patients with high cardiovascular risk and diffuse atherosclerosis [10]. Hypertension has been reported to be the most common morbidity encountered in patients with cancer (37%) [11]. MM patients are known to be vulnerable to cardiovascular complications, including hypertension and heart failure, as a result of individual risk factors (age and predisposing comorbidities) and myeloma-related effects [12]. Ler is highly lipophilic and has fewer adverse effects [9] than the first or second generation DHPs (e.g., nicardipine and nifedipine.), which have short half-life times and various adverse effects [13]. Importantly, several compounds of the 1,4-DHP class have demonstrated anti-cancer effects in many cancer types [14,15]. Therefore, we hypothesized that a combined regimen of Btz and Ler may improve the efficacy and tolerability of PI-based cancer therapy and broaden the applicability of PI to solid tumors. Interestingly, we found that a combination of Btz and Ler (Btz/Ler) kills cancer cells mainly by inducing paraptosis, which is a programmed cell death mode accompanied by extensive vacuolation derived from mitochondria and the endoplasmic reticulum (ER) [16,17]. Paraptosis lacks apoptotic features, including chromatin condensation, DNA fragmentation, apoptotic body formation, and caspase dependency, and is known to require de novo protein synthesis [16,17,18]. Although the molecular basis of paraptosis still remains to be clarified, the paraptotic process is known to be associated with the perturbation of cellular proteostasis via proteasomal inhibition [18,19,20,21,22] and disruption of Ca^2+^ homeostasis [22,23,24,25]. 

Here, we show that Ler and Btz synergistically kill breast cancer cells by inducing increased ER stress and intracellular Ca^2+^ imbalance. In this process, mitochondrial Ca^2+^ accumulation critically contributes to mitochondrial dilation, mitochondrial membrane potential (MMP) loss, and subsequent ER dilation, resulting in the induction of paraptosis-associated cell death. 

## 2. Results

### 2.1. Lercanidipine Effectively Enhances PI-Mediated Cell Death in Various Cancer Cells 

In an attempt to identify a sensitizer that can effectively overcome the resistance of cancer cells to proteasome inhibitors (PIs), we investigated whether the antihypertensive drug, lercanidipine (Ler), could sensitize cancer cells to bortezomib (Btz). While treatment with Ler up to 15 μM was not notably cytotoxic toward MDA-MB 435S (breast cancer), SNU-475 (liver cancer), SNU-668 (stomach cancer), NCI-H460 (lung cancer), BxPC-3 (pancreatic cancer), or RPMI-8226 (MM) cells, Ler dose-dependently enhanced cell death in these cell lines when combined with subtoxic doses of Btz (Figure 1A). Indeed, isobologram analysis revealed that Btz and Ler synergistically induced cell death in these cancer cells (Figure 1B). We further tested whether Ler affected the viabilities of cancer cells treated with two other FDA-approved PIs, Cfz, and Ixz, and found that Ler effectively enhanced Cfz- and Ixz-mediated cell death in both MDA-MB 435S and SNU-475 cells (Figure 1C), showing synergistic effects (Figure 1D). Interestingly, Ler did not increase the death of MCF-10A (normal breast) and Chang (normal liver) cells treated with Btz, Cfz, or Ixz (Figure 1E). However, Ler effectively enhanced Cfz- or Ixz-mediated cell death in SNU-668 and BxPC-3 cells (Appendix A). Taken together, these results suggest that PI plus Ler may be preferentially cytotoxic to cancer cells, at least in the tested types of cancer cells.

As Ler belongs to the 1,4-dihydropyridine (DHP) class of calcium channel blockers [8,9], we further investigated whether other DHPs could sensitize cancer cells to Btz. We found that amlodipine (Amlo), niguldipine (Nigul), nicardipine (Nicar), and felodipine (Felo) also dose-dependently enhanced the cell death of MDA-MB 435S or SNU-475 cells when combined with subtoxic doses of Btz (Figure 2A,D). Btz and each of the other tested DHPs demonstrated synergism in these cells (Figure 2B,E), although to a lesser degree than seen in MDA-MB 435S cells treated with the combination of Btz and Ler (Btz/Ler) (Figure 1B). In contrast to the effect of Btz/Ler, which demonstrated minimal cytotoxicity in MCF-10A and Chang cells, the combinations of Btz and each of the other tested DHPs slightly reduced the viability of MCF-10A cells (Figure 2C) but not Chang cells (Figure 2F). When we further examined the effect of Btz and the other DHPs on other types of cancer cells, we found that Btz/Amlo, Btz/Nigul, Btz/Nicar, and Btz/Felo induced cell death in SNU-668, NCI-H460, and BxPC-3 cells (Appendix A), but with less synergism than seen with Btz/Ler (Figure 1B and Appendix A). These results suggest that DHPs may overcome the resistance of cancer cells to various PIs and that among the various tested combinations of PIs and DHPs, Btz/Ler may offer advantages in both safety and effectiveness.

### 2.2. Combination of Ler and Btz Induces Paraptosis in Cancer Cells

To understand how Ler overcomes the resistance of cancer cells to a PI, we first observed cellular morphologies following treatment with Btz and/or Ler. While treatment of MDA-MB 435S cells with 4 nM Btz or 10 μM Ler for 24 h did not induce any noticeable morphological change, Btz/Ler induced marked vacuolation and cell death (Figure 3A). In contrast, the same treatments did not induce any vacuolation or cell death in MCF-10A cells. The morphology of SNU-475 cells was not affected by treatment with 20 nM Btz or 10 μM Ler alone for 24 h, but notable vacuolation and cell death were induced by Btz/Ler (Figure 3B). The morphology of Chang cells was not altered by Btz and/or Ler (Figure 3B). Dramatic vacuolation and cell death were observed in SNU-668, NCI-H460, and BxPC-3 cells treated with Btz/Ler, but not in the same cells treated with either drug alone (Appendix A). When we further tested the effects of Ler and other PIs in combination, we found that extensive vacuolation and subsequent cell death were induced by Cfz/Ler, Ixa/Ler, Btz/Amlo, Btz/Nicar, Btz/Nigul, and Btz/Felo in MDA-MB 435S and SNU-475 cells, but not in MCF-10A or Chang cells (Figure 3C). These results suggest that the combination of a PI with a DHP commonly induces vacuolation-associated cell death in these cancer cells, sparing normal cells, although Btz/Ler has the most prominent cancer-selective cytotoxicity. Since apoptotic morphologies, including blebbing and apoptotic bodies, were not observed in these cancer cells following treatment with Btz/Ler, we further examined the changes in the expression of caspase-3. We found that treatment with doxorubicin (an apoptotic inducer) triggered the cleavage of caspase-3 in MDA-MB 435S cells, whereas Btz/Ler did not (Figure 3D). Btz/Ler-induced cell death and vacuolation were not blocked by the pan-caspase inhibitor, z-VAD-fmk (Figure 3E,G), supporting the idea that apoptosis is not critically involved in this cell death. Furthermore, a necroptosis inhibitor (necrostatin-1), a ferroptosis inhibitor (ferrostatin-1), and two autophagy inhibitors (3-methyladenine and bafilomycin A1) all failed to block Btz/Ler-induced cell death and vacuolation (Figure 3E,G). Although Btz treatment increased the protein levels of both LC3B-II (an autophagy marker) and p62 (a substrate of autophagy), the Btz-mediated upregulations of LC3B-II and p62 were not affected by Ler co-treatment (Figure 3F), indicating that the sensitizing effect of Ler on Btz-mediated cell death is not associated with autophagy. Taken together, these results suggest that the anti-cancer effects induced by Btz/Ler could involve an alternative cell death mode that is not associated with apoptosis, necroptosis, ferroptosis, or autophagy.

We thus investigated whether Btz/Ler-induced vacuolation is derived from changes in organelles, such as the ER and/or mitochondria. To observe these structures, we performed confocal microscopy in YFP-ER cells (MDA-MB 435S sublines stably transfected with the YFP-ER plasmid) after MitoTracker-Red (MTR) staining. In untreated cells, filamentous mitochondria and reticular-morphology ER were observed (Figure 4A). Treatment with 4 nM Btz did not markedly alter these morphologies, although it did slightly reduce the length of mitochondria. In cells treated with 10 μM Ler alone, we did not observe any noticeable change in the ER; the length of mitochondria was shortened at 8 h and mitochondria were slightly dilated at 16 h, but their filamentous morphology was restored at 24 h. In contrast, Btz/Ler-treated cells exhibited slight mitochondrial dilation from 8 h and a progressive increase in the size of the dilated mitochondria to 16 h. Mitochondrial fluorescence was weakened from 16 h of Btz/Ler treatment, possibly due to the loss of mitochondrial membrane potential (MMP). After 24 h of Btz/Ler treatment, the initiation of death-associated cellular detachment was often observed. To further investigate the relationship between mitochondrial morphology and MMP following Btz/Ler treatment, we performed confocal microscopy in YFP-Mito cells treated with Btz and/or Ler and further stained with tetramethylrhodamine methyl ester (TMRM) (Figure 4B). We found that mitochondrial dilation exhibited a peak at 16 h of Btz/Ler treatment. Interestingly, some dilated mitochondria maintained their MMP (yellow mitochondria, white arrowhead), while others lost their MMP (green mitochondria, blue arrow). By 24 h of Btz/Ler treatment, most mitochondria demonstrated irregularly fragmented morphologies; moreover, much weaker TMRM fluorescence was observed, indicating MMP loss. Following the dilation of mitochondria, dilation of the ER was also observed from 12 h of Btz/Ler treatment. The size of the dilated ER was further increased at 16 h and maintained by 24 h. 

Dilation of mitochondria and the ER is a morphological feature of paraptosis [16,17,18] and paraptosis is known to require de novo protein synthesis [16,17]. Thus, we tested the effect of the protein synthesis blocker, cycloheximide (CHX), on the Btz/Ler-induced vacuolation and cell death in these cancer cells. We found that CHX pretreatment very effectively inhibited cell death (Figure 4C) and vacuolation (Figure 4D) induced by Btz/Ler in MDA-MB 435S cells. In addition, CHX pretreatment markedly inhibited the dilation of mitochondria and the ER in YFP-Mito and YFP-ER cells treated with Btz/Ler for 12 h (Figure 4E). However, z-VAD, necrostatin-1, ferrostatin-1, 3-MA, or bafilomycin A1 had no effect on the dilation of mitochondria and the ER (Appendix A). Collectively, these results suggest that Btz/Ler kills various cancer cells by inducing paraptosis-associated cell death.

### 2.3. Combination of Btz and Ler Enhances ER Stress

The major mechanism of PI-induced cell death induction involves the accumulation of toxic poly-ubiquitinated proteins and misfolded protein aggregates, which induce ER stress [26]. Therefore, we investigated whether the underlying mechanism of Btz/Ler-mediated anticancer effect was associated with ER stress. Western blotting revealed that Ler markedly enhanced the Btz-mediated accumulation of poly-ubiquitinated proteins (Figure 5A). Immunocytochemistry of ubiquitin showed that while ubiquitinated proteins formed aggresome-like structures in MDA-MB 435S cells treated with Btz for 16 h, more scattered expression of ubiquitinated protein aggregates was detected in MDA-MB 435S cells treated with Btz/Ler for the same period (Figure 5B). When we examined the expression levels of ER stress marker proteins, we found that treatment with Btz alone for 8 h very slightly increased the levels of phosphorylated PERK, phosphorylated eIF2α, ATF4, and CHOP (Figure 5A). While Ler alone had no effect on these proteins, Ler co-treatment markedly increased the small Btz-mediated upregulation of these ER stress marker proteins. At 16 h of treatment, Btz/Ler-induced phosphorylation of PERK and eIF2α was further increased, compared to that at 8 h, although the protein levels of ATF and CHOP were not further increased. Taken together, these results suggest that Ler may sensitize cancer cells to Btz-mediated anti-cancer effects by promoting the accumulation of misfolded proteins within the ER, thereby contributing to proteotoxic ER stress and ER dilation. 

### 2.4. Mitochondrial Ca^2+^ Overload Induced by Btz and Ler is Critical for Mitochondrial Dilation during The Progression of Paraptosis

We previously reported that disruption of Ca^2+^ homeostasis, particularly mitochondrial Ca^2+^ overload, is critical for curcumin- or celastrol-induced paraptosis, especially in the context of mitochondrial dilation [22,23]. Therefore, we investigated the possible involvement of mitochondrial Ca^2+^ imbalance in Btz/Ler-induced paraptosis. We first tested the effect of Btz and/or Ler on mitochondrial Ca^2+^ levels. Indeed, confocal microscopy and flow cytometry using Rhod-2 (a fluorescent dye that detects mitochondrial Ca^2+^) revealed that treating MDA-MB 435S cells with Btz weakly increased mitochondrial Ca^2+^ levels at 12 h. Combinations of Btz/Ler markedly increased mitochondrial Ca^2+^ levels beginning at 4 h, with a peak at 12 h (Figure 6A,B). We next investigated the functional significance of mitochondrial Ca^2+^ overload in Btz/Ler-induced cell death. MCU is known to drive a rapid and massive entry of Ca^2+^ into mitochondria at high Ca^2+^ concentrations consistent with those found at microdomains called mitochondria-associated membranes (MAMs) [27,28,29] We found that pretreatment of MDA-MB 435S cells with the MCU inhibitors, Ru360 and ruthenium red (RR), very effectively blocked Btz/Ler-induced cell death (Figure 6C). Additionally, the Btz/Ler-induced dilations of mitochondria and the ER were very markedly inhibited by Ru360 (Figure 6D). These results suggest that MCU-mediated mitochondrial Ca^2+^ overload plays an important role as an initial signal for Btz/Ler-induced paraptosis. Next, we investigated the source of the Ca^2+^ that leads to mitochondrial Ca^2+^ overload during Btz/Ler-induced paraptosis. Mitochondria are juxtaposed to the ER, a major Ca^2+^ reservoir in the cells, and the release of Ca^2+^ from the ER is mainly mediated by IP_3_ receptors (IP_3_Rs) and ryanodine receptors (RyRs) [30]. Therefore, we investigated whether Btz/Ler-induced mitochondrial accumulation is affected by pretreatment with 2-APB (an IP_3_R antagonist) or dantrolene (a RyR antagonist). We found that dantrolene significantly inhibited Btz/Ler-induced cell death, although its blocking effect was less than that of RR or Ru360 (Figure 6C). 2-APB pretreatment had no effect (Figure 6C). While Ru360 pretreatment almost completely blocked mitochondrial Ca^2+^ accumulation in Btz/Ler-treated cells, dantrolene pretreatment markedly attenuated this outcome (Figure 6D). Pretreatment with 2-APB had no effect on Btz/Ler-induced mitochondrial Ca^2+^ accumulation (Figure 6D). In addition, Ru360 pretreatment very effectively blocked the Btz/Ler-induced dilations of mitochondria and the ER, whereas dantrolene pretreatment markedly, but not completely, blocked these outcomes (Figure 6E). Taken together, these results suggest that RyR-mediated Ca^2+^ release from the ER partially, yet critically, contributes to MCU-mediated mitochondrial Ca^2+^ accumulation and affects subsequent ER dilation during the progression of Btz/Ler-induced paraptosis.

Ler, a L-type Ca^2+^ channel blocker, is known to reduce intracellular Ca^2+^ levels [8,9]. However, since Btz/Ler dramatically increased mitochondrial Ca^2+^ levels, we also examined the effects of Btz and/or Ler on cytosolic Ca^2+^ levels. Interestingly, both fluorescence microscopy and flow cytometry using Fluo-3 (a fluorescent dye that detects cytosolic Ca^2+^) showed that Ler alone weakly increased intracellular Ca^2+^ levels beginning at 4 h (about 1.7 fold) and that Btz alone slightly increased these levels beginning at 16 h (about 1.8 fold) (Figure 7A,B). Btz/Ler together notably increased intracellular Ca^2+^ levels beginning at 12 h and enhanced these levels by 3 fold at 24 h. These results indicate that Btz/Ler-induced mitochondrial Ca^2+^ overload precedes the increase in cytosolic Ca^2+^ levels (Figure 6B and Figure 7B). BAPTA-AM pretreatment partially inhibited Btz/Ler-induced cell death (Figure 7C). Interestingly, BAPTA-AM pretreatment and effectively blocked the dilation of the ER, but not mitochondria, during this process (Figure 7D). These results suggest that mitochondrial Ca^2+^ accumulation is required for mitochondrial dilation during the early phase of Btz/Ler treatment, and contributes to subsequent ER dilation. The increase in cytosolic Ca^2+^ levels seen during the late phase of Btz/Ler treatment may be important for ER dilation.

### 2.5. Combination of Btz and Ler Aggravates ER Stress and Disrupts Ca^2+^ Homeostasis Selectively in Breast Cancer Cells

Since Btz/Ler appeared to induce paraptosis selectively in cancer cells, we next investigated whether Btz/Ler differentially modulates the key signals of paraptosis in cancer cells versus normal cells. Comparing the expression of ER stress-related proteins revealed that poly-ubiquitinated proteins and eIF2α phosphorylation were slightly upregulated in MDA-MB 435S and MCF-10A cells treated with Btz alone for 8 h, and that Ler co-treatment further enhanced these levels in MDA-MB 435S cells, but not in MCF-10A cells (Figure 8A). The basal protein levels of PERK were much lower in MCF-10A cells than in MDA-MB 435S cells, and the Btz/Ler-induced enhancement of PERK phosphorylation that was noted in MDA-MB 435S cells was not observed in MCF-10A cells. Moreover, a marked CHOP upregulation was observed in MDA-MB 435S cells treated with Btz/Ler, but not in MCF-10A cells. These results suggest that Btz/Ler may modulate the unfolded protein response (UPR) differentially in these breast cancer cells versus normal breast cells. Consistent with this, Btz/Ler markedly increased mitochondrial and cytosolic Ca^2+^ levels in MDA-MB 435S cells, but not in MCF-10A cells (Figure 8B,C). Taken together, these results suggest that the apparently preferential cytotoxic effect of Btz/Ler in cancer cells may be associated with the cancer-selective aggravation of ER stress and disruption of Ca^2+^ homeostasis.

### 2.6. Mitochondrial Ca^2+^ Overload Critically Contributes to The Dilation of Both Mitochondria and The ER in Btz/Ler-Induced Paraptosis

Among the tested inhibitors in our study, CHX and Ru360 demonstrated the most potent blocking effects on Btz/Ler-induced cell death as well as the dilation of both mitochondria. Therefore, we further investigated the relationship between the key signals involved in Btz/Ler-induced paraptosis using them. We found that CHX pretreatment very effectively inhibited all the key signals of Btz/Ler-induced paraptosis, including mitochondrial Ca^2+^ accumulation (Figure 9A,B), MMP loss (Figure 9C), upregulations of the polyubiquitinated proteins, ATF4, and CHOP (Figure 9D), and the increase in cytosolic Ca^2+^ levels (Figure 9E). These results suggest that blocking of de novo protein synthesis prevents all the cellular responses to Btz/Ler. Ru360 pretreatment also markedly inhibited Btz/Ler-induced mitochondrial Ca^2+^ accumulation (Figure 9A,B), MMP loss (Figure 9C), and the increase in cytosolic Ca^2+^ levels (Figure 9E), but partially upregulations of the poly-ubiquitinated proteins, ATF4 and CHOP (Figure 9D). These results suggest that mitochondrial Ca^2+^ overload (which is important for mitochondrial dilation) may contribute to the increase in the cytosolic Ca^2+^ levels (which is important for ER dilation) in Btz/Ler-induced paraptotic signaling, although it partially affects the enhancement of ER stress. As the cause of the delayed increase in cytosolic Ca^2+^ levels, we cannot exclude the possibility that excessive overloading of Ca^2+^ in mitochondria may open the mitochondrial permeability transition pore (mPTP), contributing to the increase in cytosolic Ca^2+^ levels at the late phase of Btz/Ler treatment (Figure 9F). In addition, the RyR-mediated release of Ca^2+^ from the ER may also contribute to the late increase in cytosolic Ca^2+^ levels. We found that BAPTA-AM, which effectively inhibited Btz/Ler-induced ER dilation but only partially inhibited cell death (Figure 7C,D), failed to block the Btz/Ler-induced upregulations of poly-ubiquitinated proteins, ATF4, and CHOP. These results suggest that the delayed increase in cytosolic Ca^2+^ levels may be critical for the ER dilation, independent of the aggravation of ER stress. 

In summary, our novel results reveal that the combination of Btz and Ler demonstrates anti-cancer effects by inducing paraptosis, in which disruption of Ca^2+^ homeostasis (particularly mitochondrial Ca^2+^ overload) and ER stress play critical roles.

## 3. Discussion

We herein show for the first time that sub-lethal doses of Ler can effectively overcome the resistance of cancer cells to PIs by inducing paraptosis-associated cell death. Ler shows greater vascular selectivity than other DHPs (e.g., Nicar, Felo, Amlo, and Nigul) [31,32]. The favorable efficacy and safety profile of Ler has made it a flexible choice for antihypertensive treatment across a broad range of patients [33]. Here, we found that Ler effectively enhances the anti-cancer effect of various PIs, including Btz, Cfz, and Ixz, in MDA-MB 435S breast cancer cells and SNU-475 hepatocellular carcinoma cells, while sparing MCF-10A normal breast cells and Chang normal liver cells. The synergistic effects of Ler and Btz were also observed in other types of cancer cells, including gastric, lung, pancreatic cancer, and multiple myeloma cells. Our observations of the morphologies in the tested cancer cells revealed that sublethal doses of Btz or Ler did not induce notable vacuolation, but the combination of the two (Btz/Ler) dramatically triggered cytoplasmic vacuolation and subsequent cell death. When we tested the combination effects of Btz and various DHPs, we found that these treatments demonstrated synergistic killing effects in MDA-MB 435S cells but they showed slight cytotoxicities in MCF-10A cells (but not in Chang cells), compared to the effect of Btz/Ler. Taken together, these results suggest that the combination of a PI and Ler may offer a safe and effective therapeutic strategy to improve the effect of PI-based cancer therapy.

When we examined the underlying mechanism by which Ler overcomes the resistance of cancer cells to Btz, we found that Ler co-treatment markedly enhanced Btz-induced increases in poly-ubiquitinated protein accumulation, PERK and eIF2α phosphorylation, and ATF4 and CHOP levels. Therefore, we speculate that treatment with Btz alone may not be enough to trigger cancer cell death, since the induction of the UPR due to proteasomal inhibition may act as an adaptive mechanism for cell survival. On the other hand, the Ler-induced potentiation of Btz-mediated ER stress may trigger the accumulation of misfolded proteins within the ER, contributing to ER dilation and subsequent cell death. Supporting our idea, ER-derived vacuolization was proposed to be associated with a massive buildup of misfolded proteins within the ER lumen, the resultant increase in osmotic force, and the consequent influx of water from the cytoplasm [34]. The cytoplasmic vacuoles seen prior to Btz/Ler-induced cancer cell death originated from not only the ER but also mitochondria. In response to Btz/Ler, mitochondrial dilation was first seen at 8 h and peaked at 16 h; thereafter, the mitochondria became shrunken. In contrast, ER dilation was first evident at 12 h and maintained until death-associated cellular detachment. These results suggest the possibility that the signals responsible for mitochondrial dilation may affect the signals responsible for ER dilation. 

We previously showed that various natural products, including curcumin [21,22] and celastrol [23], effectively kill cancer cells by inducing paraptosis. In those studies, we showed that proteasome inhibition is necessary but not sufficient for this process and that disruption of Ca^2+^ homeostasis, particularly mitochondrial Ca^2+^ influx, critically contributes to the effective induction of paraptosis. In the present study, we found that Btz/Ler induced the accumulation of Ca^2+^ in mitochondria and a consequent increase in cytosolic Ca^2+^ levels, and that blocking the MCU-mediated mitochondrial Ca^2+^ influx inhibited Btz/Ler-induced mitochondrial dilation, ER dilation, and subsequent cell death. Interestingly, pretreatment with MCU inhibitors blocked the ability of Btz/Ler to trigger MMP loss and increase cytosolic Ca^2+^ levels. Scavenging of intracellular Ca^2+^ levels using BAPTA-AM effectively blocked Btz/Ler-induced ER dilation but not mitochondrial dilation. These results suggest that MCU-mediated mitochondrial Ca^2+^ overload may play a critical role in Btz/Ler-induced mitochondrial dilation, subsequent ER dilation, and cell death, whereas increased cytosolic Ca^2+^ at the late phase of Btz/Ler treatment may be important for ER dilation. Considering that the loss of MMP was accompanied by dramatic mitochondrial dilation and subsequent deformation of mitochondria at the late phase of Btz/Ler treatment, we presume that progressive loading of Ca^2+^ to mitochondria may cause mitochondrial dilation and prompt the mitochondrial permeability transition pore (mPTP) to open [35,36,37], resulting in the leakage of mitochondrial Ca^2+^ into the cytosol [37,38,39]. Supporting our idea, the application of menadione (vitamin K3 prodrug) or the ER stress inducers (such as A23187, thapsigargin, and tunicamycin), were shown to induce the influx of Ca^2+^ into the mitochondrial matrix, yield a protracted elevation of matrix Ca^2+^ beyond a critical threshold, and consequently trigger mitochondrial membrane permeabilization [36,37]. In addition, elevated Ca^2+^ in mitochondria was shown to induce the release of Ca^2+^ into the cytosol through the mPTP opening [40]. However, additional work is needed to clarify the underlying mechanism by which increased cytosolic Ca^2+^ contributes to ER dilation. Interestingly, the Btz/Ler-induced upregulation of the ER stress marker proteins was blocked almost completely by CHX, partially blocked by Ru360, but not blocked by BAPTA-AM. This seemed to follow the sequential time-course expression patterns of ER stress marker proteins, mitochondrial Ca^2+^ overload, and the increase in cytosolic Ca^2+^ levels. There may also be some crosstalk between mitochondrial Ca^2+^ overload and the aggravation of ER stress in Btz/Ler-induced signaling. However, the increase in cytosolic Ca^2+^ levels clearly acts downstream of mitochondrial Ca^2+^ overload in this signaling and does not directly affect the modulation of ER stress (Figure 9F). Further work is also needed to determine whether there is a connection between the accumulation of misfolded proteins within the ER and the increase in cytosolic Ca^2+^ levels or if these two mechanisms independently contribute to the ER dilation. 

In the present study, Btz/Ler increased both mitochondrial and cytosolic Ca^2+^ levels, in spite of the fact that Ler, an L-type Ca^2+^ channel blocker, is known to reduce intracellular Ca^2+^ levels [8,9]. When we examined the source of Ca^2+^ to trigger MCU-mediated Ca^2+^ uptake in mitochondria, we found that RyR in the ER may be an initial and key mediator in Btz/Ler-induced paraptosis. Pretreatment with dantrolene (a RyR antagonist) but not 2-APB (an IP_3_R antagonist) markedly inhibited Btz/Ler-induced mitochondrial Ca^2+^ accumulation, dilation of mitochondria and the ER, and subsequent cell death. Consistent with our results, Weigl et al. showed that DHPs, including nifedipine, can induce the release of Ca^2+^ from ryanodine-sensitive stores via a mechanism involving the DHP receptor (DHPR) and RyR [41]. The authors proposed that binding of DHP to DHPR triggers conformational changes in DHPR, leading to its stabilization in the closed-channel state, which is sufficient to activate RyR [42,43,44]. In addition, proteasome and calpain were previously reported to mediate the degradation of RyR [45]. Therefore, we speculate the possibility that both Ler and Btz may stabilize DHPR and RyR, leading to the release of Ca^2+^ from the ER and subsequent MCU-mediated Ca^2+^ uptake into mitochondria. Regarding the cause of the less potent blocking effects of dantrolene on Btz/Ler-induced paraptotic events, compared to the effects of Ru360 or RR, we presume that a yet-unidentified Ca^2+^ release channel, other than RyR or IP_3_Rs, may be also associated with the MCU-mediated influx of Ca^2+^ seen following Btz/Ler treatment. Taken together, our results suggest that Ler aggravates Btz-mediated ER stress and ER vacuolation by perturbing Ca^2+^ homeostasis, particularly through mitochondrial Ca^2+^ overload. Interestingly, Btz/Ler-induced cell death accompanied by vacuolation was not observed in the tested normal cells, including MCF-10A (breast) and Chang (liver) cells. When we examined the effects of Btz/Ler on the UPR, we found that Btz/Ler markedly increased PERK phosphorylation and CHOP expression in MDA-MB 435S cells, but not in MCF-10A cells. In addition, Btz/Ler-induced increases in mitochondrial and cytosolic Ca^2+^ levels were observed in MDA-MB 435S cells, but not in MCF-10A cells. These results suggest that the preferential cytotoxicity of Btz/Ler in our tested cancer cells may be associated with aggravation of ER stress and disruption of Ca^2+^ homeostasis in cancer cells. Aneuploidy and rapid protein synthesis of tumor cells may lead to higher ER stress than normal cells [18] and this state of ER stress may make tumor cells more vulnerable to Btz/Ler-induced impairment of proteostasis and Ca^2+^ homeostasis than normal cells.

A combined regimen of a PI plus Ler may have many advantages, including strong therapeutic effectiveness, better safety (e.g., the use of a lower PI dose could minimize toxicity), the potential to expand the applicability of PIs to solid tumors, and the opportunity to lower the therapeutic cost by using a smaller dose of the expensive PI along with Ler, which is relatively inexpensive (~2 US dollar/20 mg). In addition, clinical trials using PIs and Ler may be relatively easy to initiate, as the safety profiles, pharmacokinetics, and metabolism of both Btz and Ler have already been established [46,47]. In the future, our present findings should be confirmed by in vivo experiments that will facilitate their successful translation to the clinic. 

## 4. Materials and Methods

### 4.1. Chemicals and Antibodies

Lercanidipine, Amlodipine, Nicardipine, Niguldipine, and Felodipine were purchased from Tocris (Tocris Bioscience, Bristol, UK). Bortezomib and Carfilzomib were from Selleckchem (Houston, TX, USA). Ixazomib (MLN9708) was purchased from Cayman Chemical (Ann Arbor, MI, USA). Necrostatin-1, 3-methyladenine (3-MA), bafilomycin A1, 2-bis(o-aminophenoxy)ethane-N, N, N’N’tetraacetic acid acetoxymethyl ester (BAPTA-AM), cycloheximide and ruthenium red (RR) were from Sigma-Aldrich (St Louis, MO, USA). Ru360 was from Calbiochem (EDM Millipore Corp., Billerica, MA, USA). MitoTracker-Red (MTR), propidium iodide (PI), Fluo-3-AM, Rhod-2-AM and tetramethylrhodamine methyl ester (TMRM) were purchased from Molecular Probes (Eugene, OR, USA). z-VAD-fmk was from R&D systems (Minneapolis, MN, USA). The primary antibody against ubiquitin (sc-8017), ATF4 (sc-200) and β-actin (sc-47778) and were from Santa Cruz Biotechnology (Santa Cruz, CA, USA). p62 (610833) was purchased from BD sciences (San Jose, CA, USA). Cleaved caspase-3 (#9661), LC3B (#2775), PERK (#5683), p-eIF2α (#9721), eIF2α (#9722), and CHOP/GADD153 (#2895) were purchased from Cell Signaling Technology (Danvers, MA, USA). Secondary antibodies (rabbit IgG HRP(G-21234) and mouse IgG HRP (G-21040)) were from Molecular Probes.

### 4.2. Cell Culture

MDA-MB 435S (breast cancer), BxPC3 (pancreatic cancer), SNU-668 (stomach cancer), NCI-H460 (lung cancer), RPMI-8226, and MCF-10A (breast epithelial) were purchased from American Type Culture Collection (ATCC, Manassas, VA, USA). MDA-MB 435S cells were cultured in DMEM and BxPC-3, NCI-H460, SNU-668, and RPMI-8226 cells were cultured in RPMI-1640 medium supplemented with 10% fetal bovine serum (FBS) and 1% antibiotics (GIBCO-BRL, Grand Island, NY, USA). MCF-10A cells were maintained in DMEM/F12 supplemented with pituitary extract, insulin, human epidermal growth factor, hydrocortisone, and choleratoxin (Calbiochem). Cells were incubated in 5% CO_2_ at 37 °C. 

### 4.3. Cell Viability Assay 

Cells were cultured in 24-well plates and treated as indicated. The cells were then fixed with methanol/acetone (1:1) at −20 °C for 5 min, washed three times with PBS, and stained with propidium iodide (PI; final concentration, 1 μg/mL) at room temperature for 10 min. The plates were imaged on an IncuCyte device (Essen Bioscience, Ann Arbor, MI, USA) and analyzed using the IncuCyte ZOOM 2016B software. The processing definition of the IncuCyte program was set to recognize attached (live) cells by their red-stained nuclei. The percentage of live cells was normalized to that found in untreated control cultures (100%). The viability of RPMI-8226 cells was assessed using a CellTiter-Glo^®^ Luminescence kit (Promega, WI, USA) according to the manufacturer’s recommendations.

### 4.4. Morphological Examination of The ER and Mitochondria

To establish the stable cell lines expressing the fluorescence specifically in the ER or mitochondria, MDA-MB 435S cells were transfected with the pEYFP-ER or pEYFP-mitochondria vector (Clontech, Mountain View, CA, USA). Stable cell lines expressing pEYFP-ER (YFP-ER) or pEYFP-mitochondria (YFP-Mito) were selected with fresh medium containing 500 μg/mL G418 (Calbiochem). After treatments, YFP-ER cells were stained with 100 nM MitoTracker-Red (MTR) or YFP-Mito cells were stained with 200 nM TMRM for 10 min and morphological changes of the ER and mitochondria were observed under a K1-Fluo confocal laser scanning microscope (Nanoscope Systems, Daejeon, Korea). 

### 4.5. Immunoblot Analyses

Immunoblot analysis was performed as described previously [21]. The fold change of each target protein level compared to β-actin was determined by densitometric analysis. The relative intensities of proteins were obtained by densitometry using ImageJ software (1.50i, National Institutes of Health, Bethesda, MD, USA) and presented relative to the level of β-actin.

### 4.6. Immunofluorescence Microscopy

After treatments, the cells were fixed with acetone/methanol (1:1) for 5 min at –20°C and blocked in 5% BSA in PBS for 30 min. Fixed cells were incubated overnight at 4 °C with primary antibodies diluted in PBS [anti-ubiquitin (1:500, mouse, Santa Cruz Biotechnology)], washed three times in PBS, and incubated for 1 h at room temperature with anti-mouse Alexa Fluor 488 (1:1000, Molecular Probes). Slides were mounted with ProLong Gold antifade mounting reagent (Molecular Probes) and cell staining was visualized with the K1-Fluo confocal laser-scanning microscope (Nanoscope Systems, Daejeon, Korea).

### 4.7. Measurement of Cytosolic and Mitochondrial Ca^2+^ Levels

To measure cytosolic [Ca^2+^]_c_ levels, treated cells were incubated with 1 μM Fluo-3-AM at 37 °C for 20 min, washed with HBSS (without Ca^2+^ or Mg^2+^), and analyzed immediately by flow cytometry using a FACSAria^TM^ III (BD Biosciences, SanJose, CA, USA), or visualized by the K1-Fluo confocal laser scanning microscope. To measure mitochondrial Ca^2+^ levels ([Ca^2+^]_mt_), treated cells were incubated with 1 μM Rhod-2-AM at 4 °C for 30 min, washed with HBSS (without Ca^2+^ or Mg^2+^), further incubated with HBSS at 37 °C for 20 min, and then analyzed by flow cytometry (FACSAria^TM^ III) or visualized by the K1-Fluo confocal laser scanning microscope.

### 4.8. Isobologram Analysis

To determine how the combinations of PIs and DHPs affected the cancer cell lines, dose-dependent effects were determined for each compound alone and with a fixed concentration of the other co-treated agent. The interactions of the PIs and DHPs were quantified by determining the combination index (CI), in accordance with the following classic isobologram equation [48]: CI = (D)_1_/(Dx)_1_ + (D)_2_/(Dx)_2_, where (Dx)_1_ and (Dx)_2_ indicate the individual doses of PIs and DHPs, respectively, required to produce an effect, and (D)_1_ and (D)_2_ are the doses of PIs and DHPs, respectively, that produce the same effect when applied in combination. From this analysis, the combined effects of the two drugs can be summarized as follows: CI < 1 indicates synergism; CI = 1 indicates summation (additive and zero interaction); CI > 1 indicates antagonism.

### 4.9. Statistical Analysis 

All data are presented as mean ± SD (standard deviation) from at least three separate experiments. To perform statistical analysis, GraphPad Prism (GraphPad Software Inc, San Diego, CA, USA) was used. The normality of data was assessed by Kolmogorov–Smirnov tests and equal variance using Bartlett’s test. For a normal distribution, statistical differences were determined using an analysis of variance (ANOVA) followed by Bonferroni multiple comparison tests. If the data were not normally distributed, Kruskal–Wallis test was performed followed by Dunn’s test.

## Figures and Tables

**Figure 1 ijms-20-06112-f001:**
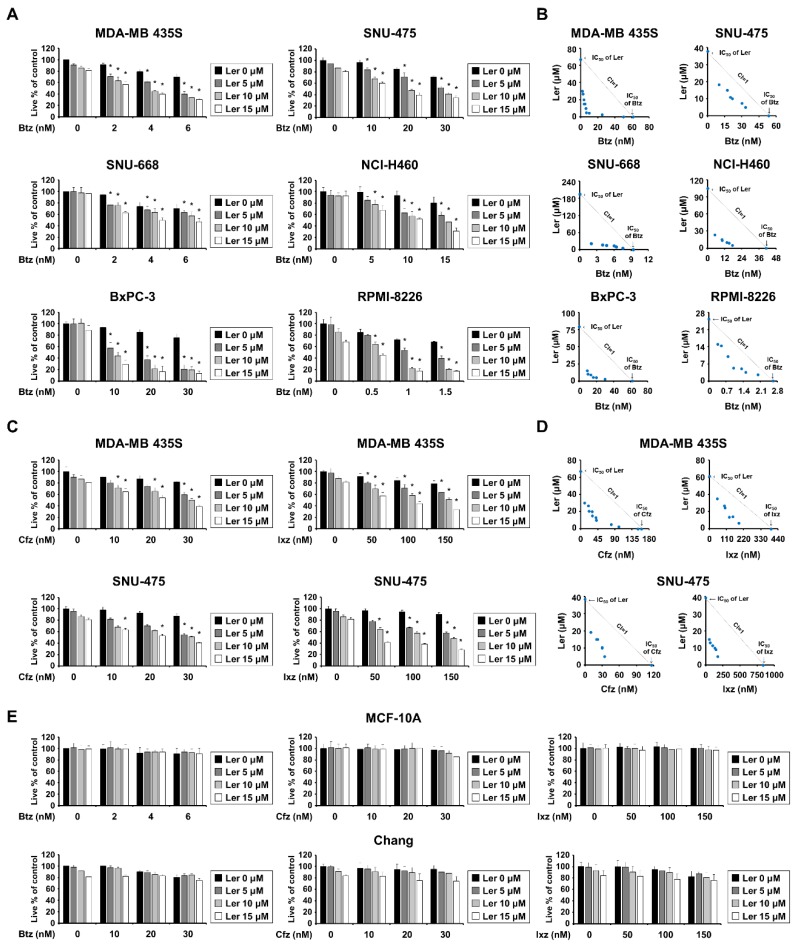
Lercanidipine (Ler) sensitizes various cancer cells, but not normal cells, to proteasome inhibitor (PI)-mediated cell death. (**A**,**C**,**E**) Cells were treated with the indicated concentrations of PIs and/or Ler for 24 h and cellular viability was assessed using the IncuCyte as described in Materials and Methods. The percentage of live cells was normalized to that of untreated control cells (100%). Data represent the means ± S.D. (*n* = 7). One-way ANOVA and Bonferroni’s post hoc test. * *p* < 0.001 vs. PI treated cells. (**B**,**D**) Isoboles for the combination of PIs and Ler that proved iso-effective (IC_50_) for inhibiting cell viability.

**Figure 2 ijms-20-06112-f002:**
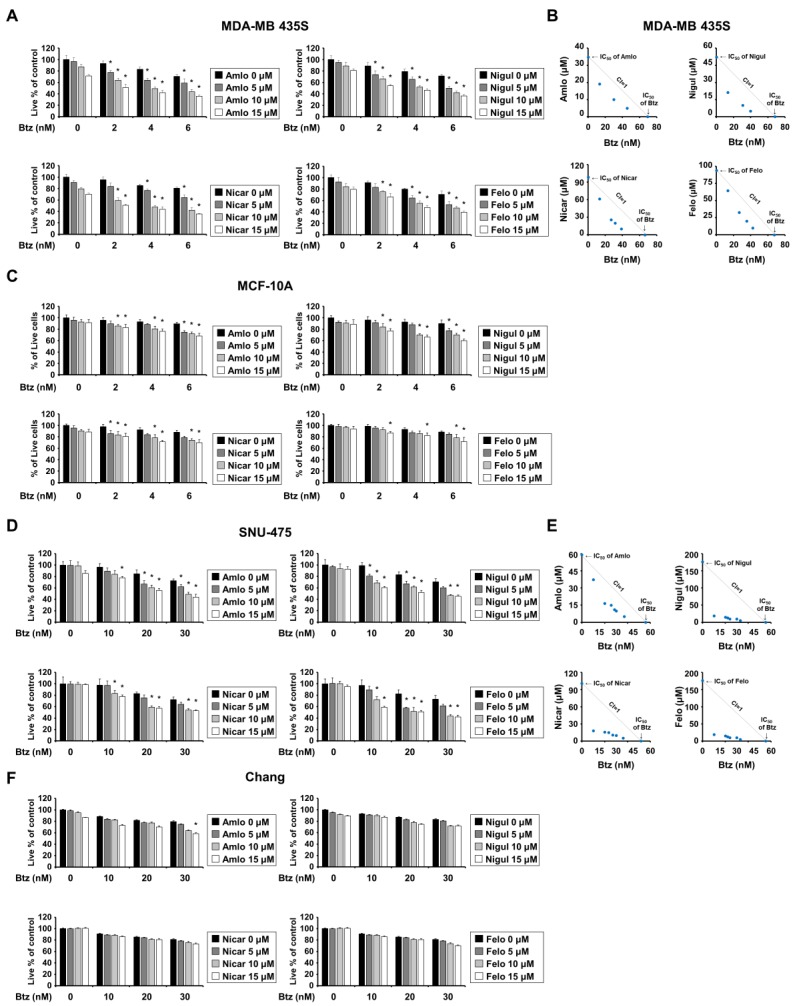
A combination of a 1,4-dihydropyridines (DHPs) and bortezomib (Btz) selectively induces cancer cell death in breast and liver cells. (**A**,**C**,**D**,**F**) Cells were treated with the indicated concentrations of Btz and/or DHPs for 24 h and cellular viability was assessed using the IncuCyte as described in Materials and Methods. The percentage of live cells was normalized to that of untreated control cells (100%). Data represent the means ± S.D. (*n* = 7). One-way ANOVA and Bonferroni’s post hoc test. * *p* < 0.001 vs. PI treated cells. (**B**,**E**) Isoboles for the combination of Btz and DHPs that proved iso-effective (IC_50_) for inhibiting cell viability.

**Figure 3 ijms-20-06112-f003:**
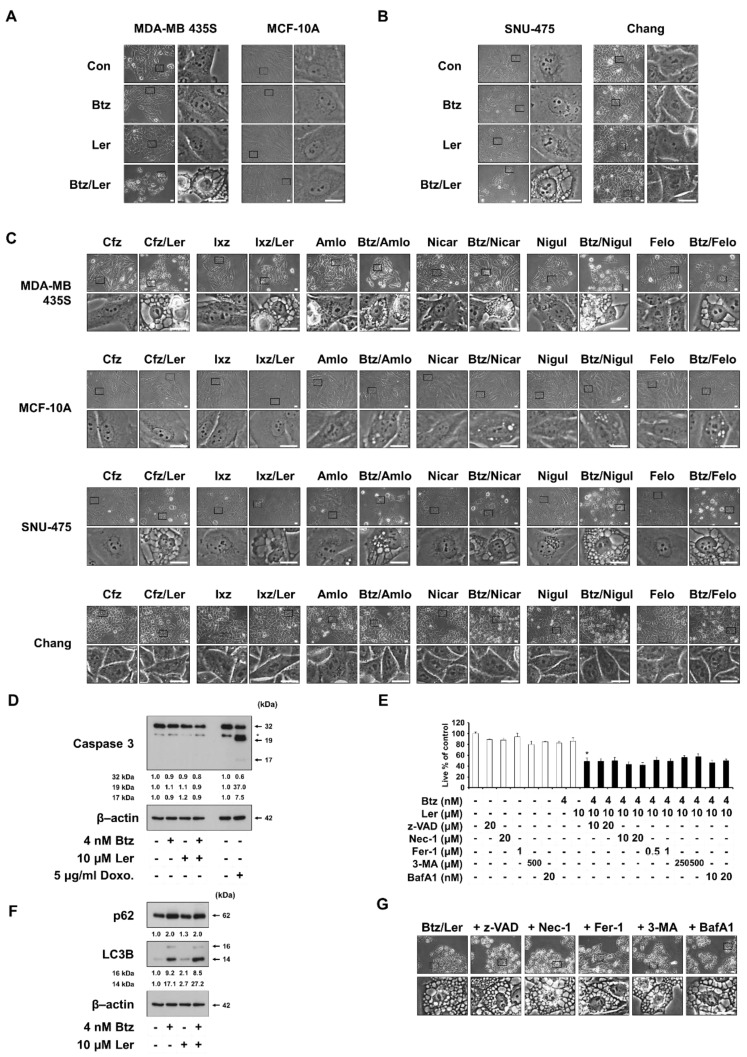
Btz/Ler induces the cancer cell death accompanied by cytoplasmic vacuolation. (**A**–**C**) Cellular morphologies were observed by phase-contrast microscopy. Bars, 20 μm. (**A**) Cells were treated with 4 nM Btz and/or 10 μM for 24 h. (**B**) Cells were treated with 20 nM Btz and/or 10 μM for 24 h. (**C**) Cells were treated with PIs (for MDA-MB 435S and MCF-10A cells, 20 nM Cfz, 100 nM Ixz, or 4 nM Btz; for SNU-475 and Chang cells, 20 nM Cfz, 100 nM Ixz, or 20 nM Btz) and/or 10 μM DHPs for 24 h. (**D**,**F**) MDA-MB 435S cells were treated with 5 μg/mL doxorubicin or 4 nM Btz and/or 10 μM Ler for 24 h. Western blotting of the indicated proteins was performed, with β-actin used as a loading control. (**E**,**G**) Cells were pretreated with the indicated inhibitors and further treated with the indicated concentrations of Btz/Ler for 24 h. (**E**) Cellular viability was assessed using the IncuCyte system. Data represent the means ± S.D. (*n* = 7). One-way ANOVA and Bonferroni’s post hoc test. * *p* < 0.001 vs. untreated cells. (**G**) Cellular morphologies were observed by phase-contrast microscopy. Bars, 20 μm.

**Figure 4 ijms-20-06112-f004:**
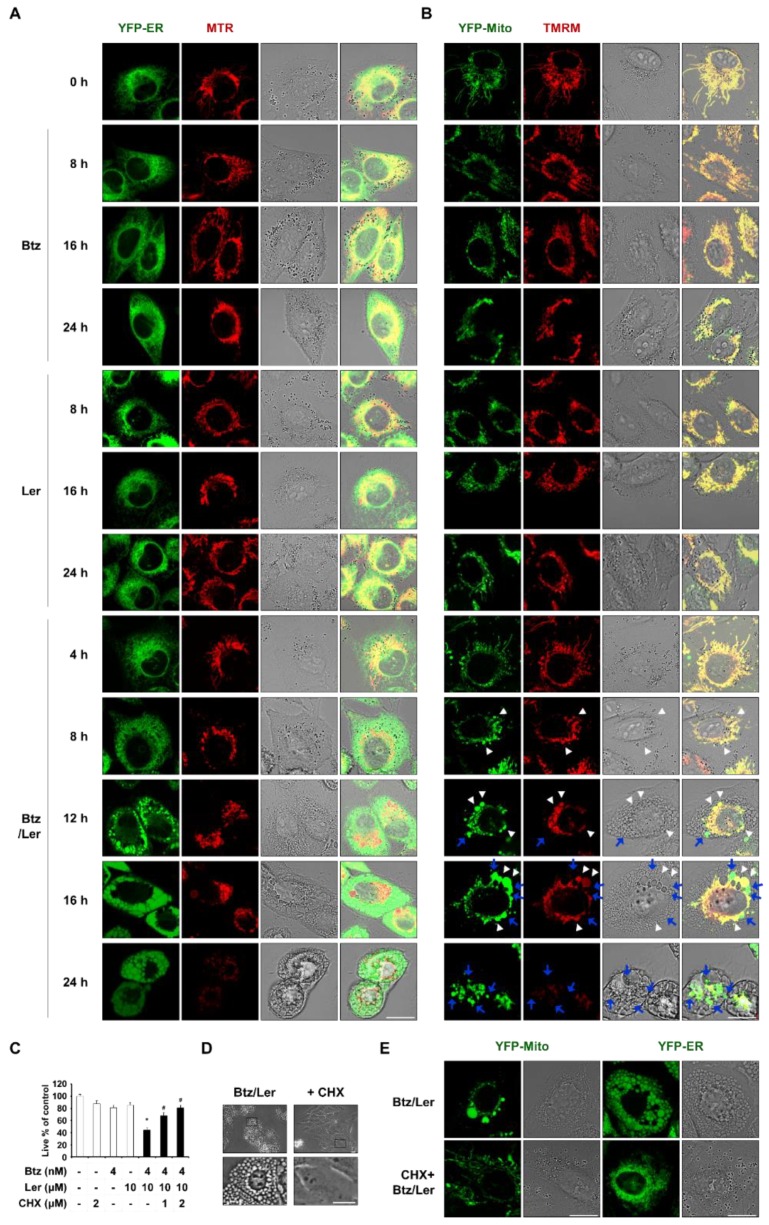
Btz/Ler induces the features of paraptosis in MDA-MB 435S cells. (**A**,**B**) YFP-ER or YFP-Mito cells were treated with 4 nM Btz and/or 10 μM Ler for the indicated time durations and then stained with MitoTracker-Red (MTR) or TMRM. Cells were observed by confocal microscopy. Bars, 20 μm. White arrowheads indicate the dilated mitochondria whose mitochondrial membrane potential (MMP) is maintained and blue arrows denote the dilated mitochondria whose MMP is lost. (**C**) Cells were pretreated with cycloheximide (CHX) and further treated with the indicated concentrations of Btz/Ler for 24 h. Cellular viability was assessed using the IncuCyte system. Data represent the means ± S.D. (*n* = 7). One-way ANOVA and Bonferroni’s post hoc test. * *p* < 0.001 vs. untreated cells; # *p* < 0.05 vs. Btz/Ler-treated cells. (**D**) MDA-MB 435S cells were untreated or pretreated with 2 μM CHX and further treated with 4 nM Btz and/or 10 μM Ler for 12 h. Cellular morphologies were observed by phase-contrast microscopy. Bars, 20 μm. (**E**) YFP-Mito and YFP-ER cells were pretreated with 2 μM CHX and further treated with 4 nM Btz plus 10 μM Ler for 12 h. Cells were observed by confocal microscopy. Bars, 20 μm.

**Figure 5 ijms-20-06112-f005:**
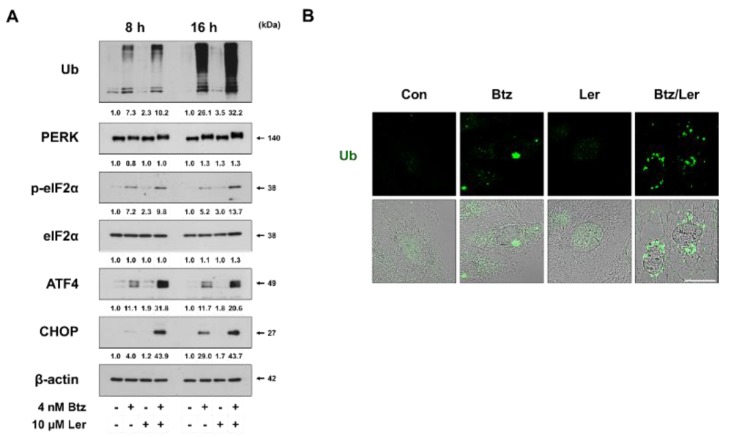
Ler enhances Btz-mediated ER stress. (**A**) MDA-MB 435S cells were treated with 4 nM Btz and/or 10 μM Ler for the indicated time durations. Western blotting of the indicated proteins was performed, with β-actin used as a loading control. (**B**) MDA-MB 435S cells were treated with 4 nM Btz and/or 10 μM Ler for 16 h. Immunocytochemistry of Ub was performed. Bar, 20 μm.

**Figure 6 ijms-20-06112-f006:**
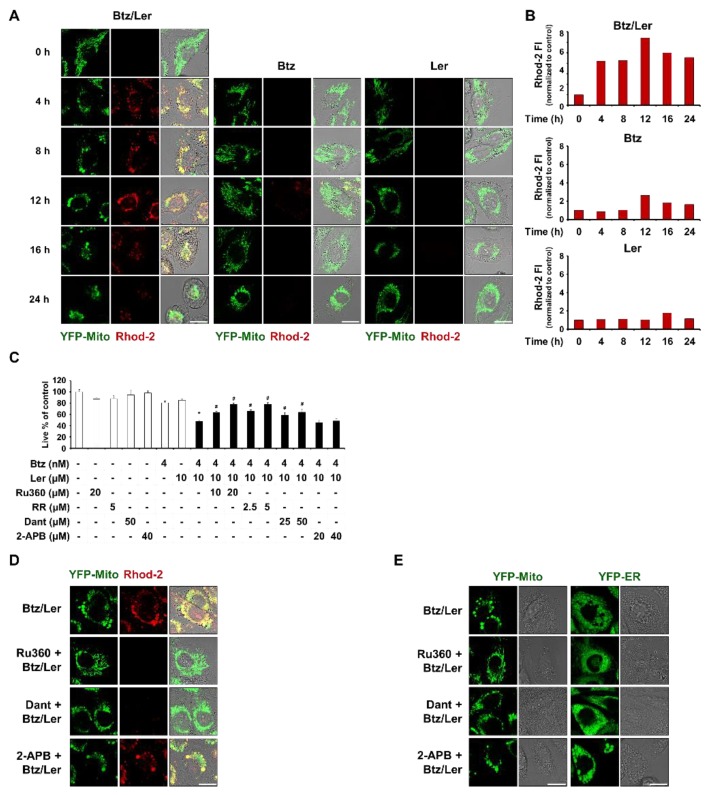
Mitochondrial Ca^2+^ overload triggers mitochondrial dilation in Btz/Ler-induced paraptosis. (**A**,**B**) Cells were treated with 4 nM Btz and/or 10 μM Ler for the indicated time points. Treated cells were stained with Rhod-2 and subjected to confocal microscopy (**A**) and flow cytometry (**B**). Bars, 20 μm. (**C**) MDA-MB 435S cells were pretreated with the indicated Ca^2+^ antagonists, and further treated with 4 nM Btz and/or 10 μM Ler for 24 h. Cellular viability was assessed using the IncuCyte as described in Materials and Methods. Data represent the means ± S.D. (*n* = 7). One-way ANOVA and Bonferroni’s post hoc test. * *p* < 0.001 vs. untreated cells; # *p* < 0.05 vs. Btz/Ler-treated cells. (**D**) YFP-Mito cells were untreated or pretreated with 20 μM Ru360, 50 μM dantrolene (Dant), or 40 μM 2-APB, and further treated with 4 nM Btz plus 10 μM Ler for 8 h. Treated cells were stained with Rhod-2 and subjected to confocal microscopy. Bars, 20 μm. (**E**) YFP-Mito and YFP-ER cells were untreated or pretreated with 20 μM Ru360, 50 μM Dant, or 40 μM 2-APB, and further treated with 4 nM Btz plus 10 μM Ler for 12 h. Treated cells were observed by confocal microscopy. Bars, 20 μm.

**Figure 7 ijms-20-06112-f007:**
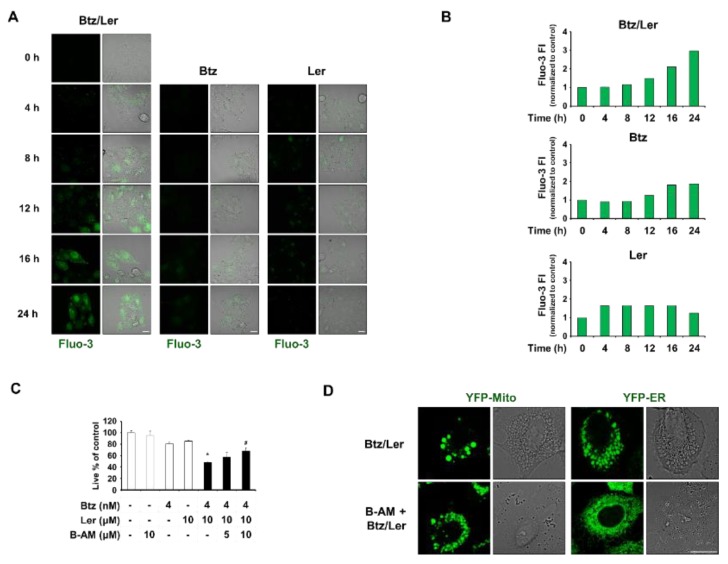
A combination of Btz and Ler induces disruption of Ca^2+^ homeostasis during paraptosis. (**A**,**B**) Cells were treated with 4 nM Btz and/or 10 μM Ler for the indicated time points. Treated cells were stained with Fluo-3 and subjected to confocal microscopy (**A**) and flow cytometry (**B**). Bars, 20 μm. (**C**) MDA-MB 435S cells were pretreated with the 10 μM BAPTA-AM, and further treated with 4 nM Btz and/or 10 μM Ler for 24 h. Cellular viability was assessed using the IncuCyte as described in Materials and Methods. Data represent the means ± S.D. (*n* = 7). One-way ANOVA and Bonferroni’s post hoc test. * *p* < 0.001 vs. untreated cells; # *p* < 0.05 vs. Btz/Ler-treated cells. (**D**) YFP-Mito and YFP-ER cells were pretreated with 10 μM BAPTA-AM and further treated with 4 nM Btz plus 10 μM Ler for 12 h. Cells were observed under the confocal microscope. Bars, 20 μm.

**Figure 8 ijms-20-06112-f008:**
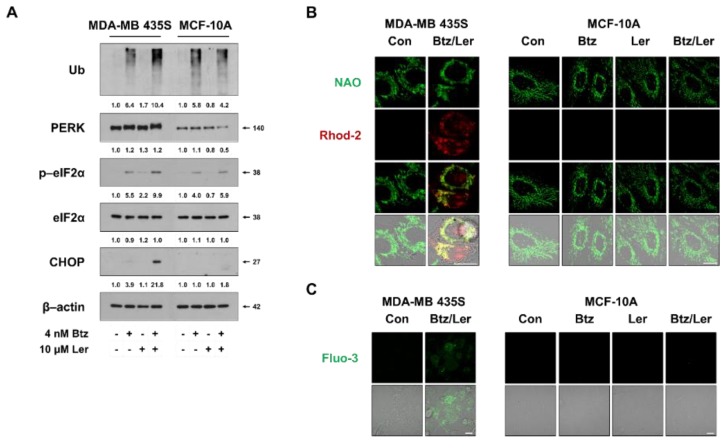
Differences in the unfolded protein response and Ca^2+^ modulation may contribute to the differential effect of Btz/Ler in cancer cells and normal cells. (**A**) Cells were treated with 4 nM Btz and/or 10 μM Ler for 8 h. Western blotting of the indicated proteins was performed, with β-actin used as a loading control. (**B**,**C**) Cells were treated with 4 nM Btz and/or 10 μM Ler for 8 h (**B**) or 24 h (**C**). Treated cells were stained with NAO/Rhod-2 (**B**) or Fluo-3 (**C**) and subjected to confocal microscopy. Bars, 20 μm.

**Figure 9 ijms-20-06112-f009:**
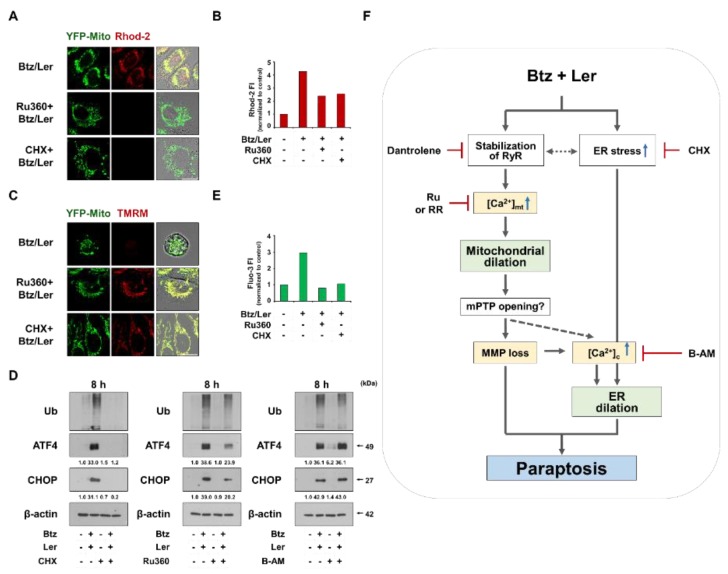
Mitochondrial Ca^2+^ overload and enhanced ER stress are critical for Btz/Ler-induced paraptosis. (**A**–**C**) YFP-Mito cells were pretreated with 20 μM Ru360 or 2 μM CHX and further treated with 4 nM Btz plus 10 μM Ler for 8 h (**A**,**B**) or 24 h (**C**). (**A**,**B**) Cells were stained with Rhod-2 and subjected to confocal microscopy (**A**) or flow cytometry (**B**). Bars, 20 μm. (**C**) Treated cells were stained with TMRM and observed by confocal microscope. Bars, 20 μm. (**D**) MDA-MB 435S cells were pretreated with 2 μM CHX, 20 μM Ru360, or 10 μM BAPTA-AM and further treated with Btz/Ler for 8 h. Western blotting of the indicated proteins was performed, with β-actin used as a loading control. (**E**) MDA-MB 435S were pretreated with 20 μM Ru360 or 2 μM CHX and further treated with Btz/Ler for 24 h. Treated cells were stained with Flou-3 and subjected to flow cytometry. (**F**) Hypothetical model of the signaling pathways involved in Btz/Ler-induced paraptosis. The blue arrows pointing up, the red “T” lines, the grey dotted arrows, and the grey arrows indicate increase, inhibition, possible regulation, and activation, respectively.

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
