# Peer review of "Lercanidipine Synergistically Enhances Bortezomib Cytotoxicity in Cancer Cells via Enhanced Endoplasmic Reticulum Stress and Mitochondrial Ca2+ Overload"

_ijms, 2019, doi:10.3390/ijms20246112_

Round 1

Reviewer 1 Report

The authors did an amazing job clarifying my concerns. They are very professional.

Author Response

I greatly appreciate the first reviewer’s compliment.

Reviewer 2 Report

In the manuscript “Lercanidipine synergistically enhances bortezomib cytotoxicity in cancer cells via enhanced endoplasmic reticulum stress and mitochondrial Ca2+ overload” Lee et al described the additive effect of Btz and Ler in a few different solid tumor models in vitro. The authors suggest Btz/Ler combination treatment induce cell death through paraptosis, a less well-known mechanism of cell death. The Authors shown lack of apoptotic markers, mitochondrial and ER vacuole formation, Ca++ overload and ER stress in cell lines treated with either single or combinational therapy. While is the manuscript is exhaustively descriptive, the study falls short on the following 4 major areas:

1. The author considered apoptosis, but not autophagy, ferroptosis both could induce mitochondrial and ER vacuole formation. If there are no good markers of paraptosis, the lack of (or not) autophagy and ferroptosis should be demonstrated. There are no rescue experiments to support authors’ claims, for example can you reverse ER stress and Ca++ overload and prevent cell death?

2. While the data clearly shows that combination treatment does not kill normal cells, but no real explanation why, based on the mechanism that the authors described in the study there shouldn’t be any differences. This should be explained in more detail in discussion.

3. Lack of in vivo experiments to show this combination therapy can be adapted to future clinical testing.

Author Response

I greatly appreciate the second reviewer’s comments. In addition, we included our detailed explanations on the effect of PI plus Ler on cancer cells versus normal cells in the Discussion of the revised manuscript. Importance of further in vivo experiments was also commented in Discussion of the Revised manuscript.

1. We already showed that the inhibitors of autophagy, including 3-MA and BafA1, did not significantly reverse the viability (Figure 3E) and vacuolation of MDA-MB 435S cells treated with Btz/Ler. To respond to this reviewer’s comment, we further examined the effects of ferrostatin-1 on Btz/Ler-induced cell death and vacuolation. Ferrotatin-1 pretreatment did not affect the viability and vacuolation in MDA-MB 435S cells treated with Btz/Ler, similar to the effects z-VAD, necrostatin-1, 3-MA and BafA1. These results using ferrostatin-1 were incorporated in Figure 3E and 3G of the revised manuscript and described as follows (line 138, page 5).

Furthermore, a necroptosis inhibitor (necrostatin-1), a ferroptosis inhibitor (ferrostatin-1), and two autophagy inhibitors (3-methyladenine and bafilomycin A1) all failed to block Btz/Ler-induced cell death (Figure 3E) and vacuolation (Figure3G). Although Btz treatment increased the protein levels of both LC3B-II (an autophagy marker) and p62 (a substrate of autophagy), the Btz-mediated upregulations of LC3B-II and p62 were not affected by Ler co-treatment (Figure 3F), indicating that the sensitizing effect of Ler on Btz-mediated cell death is not associated with autophagy. Taken together, these results suggest that the anti-cancer effects induced by Btz/Ler could involve an alternative cell death mode that is not associated with apoptosis, necroptosis, ferroptosis or autophagy.

In addition, we found that Btz/Ler-induced dilation of the ER and mitochondria was not affected by z-VAD, necrostatin-1, ferrostatin-1, 3-MA and bafilomycin A1. These results are included as Figure S4 of the revised manuscript and described as follows (line 188, page 7):

However, z-VAD, necrostatin-1, ferrostatin-1, 3-MA, or BafA1 had no effect on the dilation of mitochondria and the ER (Figure S4).

 The fact that inhibition of autophagy or ferroptosis had no effect on Btz/Ler-induced cell death as well as the dilation of the ER and mitochondria suggests that autophagy or ferroptosis may not be critically involved in the signals associated with Btz/Ler-induced cell death. Therefore, further examination of the effects of ferrostatin-1, 3-MA, and BafA1 on Btz/Ler-induced ER stress and Ca2+ overload seems to be unnecessary.

2. In Figure 8 of our previously revised manuscript, we already included the data on the differences in the UPR and Ca2+ modulation in cancer cells and normal cells and described these results as follows (line 297, page 12): 

2.5. Combination of Btz and Ler aggravates ER stress and disrupts Ca2+ homeostasis selectively in breast cancer cells.

Since Btz/Ler appeared to induce paraptosis selectively in cancer cells, we next investigated whether Btz/Ler differentially modulates the key signals of paraptosis in cancer cells versus normal cells. Comparing the expression of ER stress-related proteins revealed that poly-ubiquitinated proteins and eIF2α phosphorylation were slightly upregulated in MDA-MB 435S and MCF-10A cells treated with Btz alone for 8 h, and that Ler co-treatment further enhanced these levels in MDA-MB 435S cells, but not in MCF-10A cells (Figure 8A). The basal protein levels of PERK were much lower in MCF-10A cells than in MDA-MB 435S cells, and the Btz/Ler-induced enhancement of PERK phosphorylation that was noted in MDA-MB 435S cells was not observed in MCF-10A cells. Moreover, a marked CHOP upregulation was observed in MDA-MB 435S cells treated with Btz/Ler, but not in MCF-10A cells. These results suggest that Btz/Ler may modulate the unfolded protein response (UPR) differentially in these breast cancer cells versus normal breast cells. Consistent with this, Btz/Ler markedly increased mitochondrial and cytosolic Ca2+ levels in MDA-MB 435S cells, but not in MCF-10A cells (Figure 8B,C). Taken together, these results suggest that the apparently preferential cytotoxic effect of Btz/Ler in cancer cells may be associated with the cancer-selective aggravation of ER stress and disruption of Ca2+ homeostasis.

In response to this reviewer’s comment, we included the description of these results in the Discussion section as follows (line 442, page 16):

Interestingly, Btz/Ler-induced cell death accompanied by vacuolation was not observed in the tested normal cells, including MCF-10A (breast) and Chang (liver) cells. When we examined the effects of Btz/Ler on the UPR, we found that Btz/Ler markedly increased PERK phosphorylation and CHOP expression in MDA-MB 435S cells, but not in MCF-10A cells. In addition, Btz/Ler-induced increases in mitochondrial and cytosolic Ca2+ levels were observed in MDA-MB 435S cells, but not in MCF-10A cells. These results suggest that the preferential cytotoxicity of Btz/Ler in our tested cancer cells may be associated with aggravation of ER stress and disruption of Ca2+ homeostasis in cancer cells. Aneuploidy and rapid protein synthesis of tumor cells may lead to higher ER stress than normal cells [18] and this state of ER stress may make tumor cells more vulnerable to Btz/Ler-induced impairment of proteostasis and Ca2+ homeostasis than normal cells.

3. We completely agree with this reviewer’s opinion. At the end of the Discussion section of the revised manuscript, we described this point as follows (line 459, page 16):

In the future, our present findings should be confirmed by in vivo experiments that will facilitate their successful translation to the clinic.

Round 2

Reviewer 2 Report

While the study itself requires further refinement with better controls, molecular biology interrogations, and more importantly in vivo experiments, they can be done in future studies and should not prevent the publication of the manuscript in its current form. 

This manuscript is a resubmission of an earlier submission. The following is a list of the peer review reports and author responses from that submission.

Round 1

Reviewer 1 Report

Overall, the manuscript is well written. The paper highlights the important potential for therapeutic intervention by using two different drugs at the same time. While the implications of the research are significant, the overall novelty of the research design is rather lacking as there are many other papers reporting synergistic interactions between Bortezomib and another drug. However, this manuscript is the first paper to demonstrate Bortezomib and Lercanidipine. Additionally, there are two concerns that should be specifically addressed. 1). Reasons as to why MM cell lines were not included in this study should be discussed. This is important as Bortezomib specifically is used to treat MM, and it may seem odd that MM cell lines were excluded from cell testing. If a compelling reason cannot be given in the discussion, these cells should to added to the experiments. 2). Other non-tumorigenic cells should be used as a normal control other than MCF-10A cells. While they are indeed non-tumorigenic, they are not normal as they were originally derived from fibrocystic tissue. Other cell lines such as HMECs, or other cells closer to normal, non-immortalized cells should be used. If they cannot be used, then a sufficient explanation should be added to the discussion.

Reviewer 2 Report

In this manuscript, Lee et al. are studying a combination therapy, using a proteasomal inhibitor (bortezomib, Btz) with a calcium channel blocker (lercanidipine, Ler), as a potential cancer treatment in solid tumors. They concluded that this combination induces paraptosis in vitro by increasing endoplasmic reticulum stress and mitochondrial calcium overload. Although the paper has a potential benefit to help developing cancer treatments, however it has some concerns that need to be addressed.

-          In 37, the author mentioned that adding a non-anticancer drug will not increase the adverse effect. This is very ambitious hypothesis and the author should be careful about this claim. In his case, calcium channel blockers (CCB), especially dihydropyridines, are associated with hypotension, increased risk of myocardial ischemia and infarction, as well as increased risk of mortality in case of heart failure. Add to that the drug-drug interactions that may be occur. In this context, does introduction of CCB is justified in every cancer patient? The author suggested that in multiple myeloma, this can reduce blood pressure. What about breast cancer, and the other types of cancer mentioned in the paper, do these patients develop high blood pressure?

-          In 45, using this sentence as a reference to what the author is doing is not relevant (see 47). There is no place in the paper where the author addressed whether Ler has increased or decreased the Btz efflux or influx. The author should at least address the potential increase in cytoplasmic Btz concentration when combined with Ler to verify if this is true.

-          In Figure 1A, it seems that Ler alone has already some effect in increasing cell death, at least with SNU-668 cells. Is there a reason why the author chose the 24h instead of 48h? It would be relevant to see the effect of Ler on the different cells after 48h, especially that drug resistance is known to develop later after treatment has shown to be beneficial. Besides, in the literature, it has already been shown that PI can inhibit significantly the highly aggressive MDA-MB-231 cells at a concentration lower than 6nM, which makes more sense to increase the treatment time to 48h.

-          In Figure 1E, how does the author explain the no effect of Ler on normal cells? The same question can be addressed to Figure 3A. The author also showed the effect of Ler + different PI inhibitors in normal breast cells, which is good, however, if this added here, then you should also show the effect of these different PI on the other mentioned cancerous cells. What about the other normal cells as control?

-          In Figure 2B, why the author didn’t include the treatment of the other cancer cells? It’s good to see if the other CCB follow the same pattern. The whole figure may also be a supplementary figure.

-          In Figure 2C, the author should give at least a hypothesis on the fact that from all the DHP he used, only Btz doesn’t affect normal cell viability. Again this should be tested on the other normal control cells.

-          In Figure 3A, why the author is using different drug concentration than the ones used in the previous figures? For instance, the viability assays were all done on a max of 6nM of Btz, while in this assay, the author is using 20nM. Do each cell respond differently? What about the effect of the 20nM treatment on the normal cells? Is it toxic? The author needs to use lots of controls in order to validate his hypothesis.

-          In Figures 3A and 3B, control images for a normal cell line to SNU-668, NCI-H460, BxPC-3 should be included. (The author only compared the MCF-10A to MDA-MB 435s).

-          In Figure 3, follow the same pattern in representing the data.

-          In Figure 3B, again the author has shown the result of cell treatment with the different DHP, however, he did not show the viability assay results of the treatment in Figure 1. A control group with Ler alone should also be added to Figure 3B.

-          The supplementary figure S1, is an important figure and should be included in the original paper. In this figure, the author was trying to prove that after Ler/Btz treatment, cells did undergo paraptosis. However, this figure alone is not enough to prove this hypothesis. The author should include more relevant tests, showing no upregulation of apoptotic markers such as caspases. The author should also use a marker for paraptosis, for example, increased prohibitin accumulation in mitochondria.

-          The supplementary S2 figure should be included in Figure 3.

-          In Figure 5 A, it is important to show the data quantification.  It seems, according to the blots, that Btz alone was able to significantly increase all ER stress markers, which does not correlate with what the author have previously shown (in the previous experiments, when used alone, Btz had no effect on viability, no effect on vacuolation). The author should also show the results on a control MCF-10A. Additionally, it seems that some ER markers are more increased with Btz or Ler alone when compared to Btz/Ler (see eIFa, PERK). It would be a very important to add more blots testing caspase and some autophagy markers to rule out the involvement of apoptosis and autophagy and validate the hypothesis of paraptosis.

-          In Figure 5 A, make sure the – and + fit to the bands.

-          In Figure 6A, can the author give an explanation on how there is an intracellular calcium increase after treatment with a CCB? Even when treated with Ler alone, the author said there is no apparent increase in calcium, however, it is obvious that there is significant increase at 16h. The author should also consider showing a quantification of these images.

-          In 196, although MCU contribute to mitochondrial calcium uptake, however, they have low affinity compared to the VDAC on the outer mitochondrial membrane. VDAC should not be ruled out, especially if there is a calcium release from the ER that can subsequently lead to mitochondrial swelling.

-          In the conclusion, the author gave a confusing hypothesis about the calcium pathways, whether it originates from mitochondria or ER.

What does the author suggest about the reported PI resistance developed in cancer cells?

The author failed to link the relationship between CCB and the increase in mitochondrial calcium. The fact of using CCB to reduce intracellular calcium is already established, it would be more relevant if the author speculated on how CCB can explain the controversial increase the intracellular calcium using CCB alone (seen in Figure 6) and subsequently the increase in mitochondrial calcium when combined with Btz in cancer cells.